# Analysis of Cross-Connected Half-Bridges Multilevel Inverter for STATCOM Application

**Yuan Li [1] and Muhammad Humayun [2,*]**

[1]    State Grid Changzhou Power Supply Company, Jiangsu Power Electronics Co Ltd.,
    Changzhou 213000, China; wjgdgsyl@sohu.com
[2]    Department of Automation, SEIEE, Shanghai Jiao Tong University, Shanghai 200240, China
*    Correspondence: mhumayun88@sjtu.edu.cn

**Abstract:**   This paper suggested a single-phase cross-connected half-bridges multilevel inverter (CCHB-MLI) topology for static synchronous compensator (STATCOM) applications. The proposed MLI structure consists of cross-connected multilevel cells connected in series with a more optimized number of devices to synthesize a higher number of voltage steps. Each cell in the structure consists of a set of switches and a DC-capacitor. Typically, when several DC-capacitors are used in an inverter, the DC voltages fluctuation occurs due to tolerance between passive element and asymmetric switch losses. A dual-loop control technique has been proposed with level-shifted pulse width modulation PWM to overcome these issues. The proposed methodology balances the DC-voltages using a proportional-integral controller by adjusting the switch duty cycle. The control method helps offset the issue of aggravated fluctuation while preserving the delivered reactive power distributed equally among the DC-capacitors at the same time. A thorough comparison is made between the proposed inverter concerning the number of components and efficiency to demonstrate the effectiveness of previous topologies. Moreover, a simulation model built in simulink and experimental results take from laboratory prototype to confirm the effectiveness of proposed structure and its control technique.

**Keywords:** cross-connected H-bridge (CCHB); multilevel inverter (MLI); phase disposition (PD-PWM); DC-capacitor balancing; static compensator

## 1. Introduction

Multilevel inverters (MLIs) technology has become an important developing field in power electronics and has now become a preferred choice for a several medium and high-power applications [1]. The rise in the output voltage steps is one major driving force behind this development. The shape of the voltage waveform approaches a sinusoidal wave if the output voltage steps increases, leading to the depletion of the harmonics in the inverter output voltage. This leads to different power inverters' performance improvements, such as high-power density, reduction of voltage stress, lower emissions of electromagnetic interference (EMI), higher efficiency, long-term reliability, and reduced switching losses [2,3].

MLIs consists of multiple DC sources (such as batteries or capacitors) and switching devices (i.e., IGCTs or IGBTs). Many strategies have been taken by scientists to strengthen the efficiency of MLIs. Most of these attempts were based on synthesizing higher AC output voltages in the form of staircases by connecting power switches to DC sources and/or DC capacitors [4]. The switch voltage rating and the operating frequency are bound for high power applications. It is an overwhelming challenge and a significant prerequisite to boost the operating frequency by decreasing the switch's power rating while still retaining viable power quality [5].

The voltage source multilevel inverter (VS-MLI) is extensively investigated in the literature for different power converter applications. Over the past few decades, VS-MLIs have been widely used in DC-AC or AC-DC conversions, motor drives, battery-powered systems, such as electric vehicle fast-charging stations, and submarine propulsion. MLI has also been considered with the increasing industrial emergence of grid-connected applications such as uninterrupted power supplies (UPS), photovoltaic (PV), static synchronous compensator (STATCOM), and wind power conversion systems [6].

The STATCOM used VS-MLI to control the grid voltage, improve the power factor, control and/or manage reactive power, and stabilize the power system. When the MLI for STATCOM is used, it should be fitted with galvanically isolated DC-capacitors, excluding any DC source. This results in eliminating the use of heavy, bulky, and costly line-side transformers. In addition, an AC inductor should be installed to help to discern the voltage between STATCOM and the grid. The use of the isolated DC capacitors in an inverter also contributes to an exacerbation of voltage fluctuations in the DC-voltage of STATCOM [7].

The DC-voltage fluctuation issue caused the following:

- nonlinear and/or reactive loads
- asymmetric switching/conduction losses produced by switches
- non-ideal passive components
- voltage and current sensors accuracy

STATCOM inverters may be narrowly split into two groups in terms of DC capacitors' voltage ratings, namely symmetric and asymmetric MLIs. When similar voltage rating capacitors are used in an inverter, such an inverter is recognized as an symmetric MLI. Non-identical voltage rating capacitors, on the other hand, result in an asymmetrical MLI [8]. In recent years, both types of topologies have been studied, and several reports have been published in the literature as a consequence. Designers try to investigate the above challenges by posing different new systems with the least number of possible switches.

With several DC sources and/or DC capacitors' contribution, contemporary, reduced switch structures of the established VS-MLI topologies are proposed in [1,4,6,8–15]. While these evolved inverters have numerous advantages over conventional inverters, using the aforementioned traditional structure of inverter requires more devices. This leads to increased circuit size, expense, and design complexity. Moreover, capacitor voltages tend to diverge, resulting in the need for voltage balancing control schemes.

Evidently, by increasing the voltage steps, the performance of MLI increases. The number of devices used in an inverter is thus increasing. Consequently, MLIs with asymmetric DC sources/capacitors are designed to accomplish a more significant number of voltage steps and reduce the number of devices [16]. In order to synthesize the higher voltage steps with lease number of active and passive components, Ounejjar and Al-Haddad suggested asymmetric packed U-cells (PUC) MLI in [17]. Packed U-cell topology shows a similarity with the configuration of the flying capacitor (FC) and cascaded H-bridge MLI due to the use of isolated DC sources. Although PUC-MLI has many advantages over conventional topologies, there are also many undesirable features of PUC topology.

The limitations in the PUC topology are the asymmetrical configuration of the inverter, which cannot made the summation of the DC voltages on the output, while an increase in the voltage levels result in the need for different voltage rating capacitors. The switch power rating is a significant problem for high-power applications that leads to limitations on the inverter's functionality, and expanding the inverter voltages is not straightforward. Asymmetric packaged U-cell topology is not possible for medium-power grid-connected applications because of these limitations. A modular structure MLI composed of a simple module or cells that can easily split the desired voltage is a potential solution [18].

To achieve the required efficiency and resolve the deficiencies described above, Andres et al. introduced a cross-connected half-bridge structure in [19]. This results in the mitigation of individual

switch stress, ease of DC voltage balancing control, and intrinsic DC fault tolerance capacity relative to certain other topologies. This topology's main benefits are that it is more practical due to its simple structure, high reliability, modularity, and efficient to any number of voltage levels. The MLI cross-connected half-bridges are well-suited for medium-power grid-connected applications based on these advantages. As for previous literature, it is essential to realize that the cross-connected half-bridge (CCHB) application did not appear as a grid-connected (STATCOM) application candidate.

MLIs-based STATCOM suffer from harmonic emission, reactive power, and voltage imbalance problems due to multiple DC-capacitors uses. To mitigate harmonics, compensate for reactive power, and regulate the voltage imbalance, different researchers have made significant contributions in this area. Numerous strategies have been developed for active voltage control, highlighting the critical contributions based on proportional resonant, deadbeat and balanced integral control, etc. [20–23]. By reducing the steady-state uncertainty related to AC signal, the proportional-resonant (PR) controller achieved dominance [20]. However, numerous demerits also exist, such as the sensitivity to slight frequency shifts, the need to handle the difficult task of tuning several resonant frequencies, and the expectation of margins of instability due to the sensors' phase shift [21]. Deadbeat control shows a good dynamic response. However, this control technique exhibits model indeterminacy sensitivity, parameter inconsistency sensitivity, and noise [22]. A two-stage control method that is a redundant switching state selection (RSS) active voltage control is given in [23], where each DC capacitor voltage is compared with the reference value. To change the modulation index to match voltages, the voltage steady-state error is compensated by the proportional and integral (PI) controller for each DC capacitor. However, in the current literature, the CCHB-MLI DC-capacitor voltage balancing solution is not discussed.

In light of the above, for the STATCOM application in the current study, a new simple modular structure of five-level cross-connected half-bridges MLI is implemented. High power efficiency and an essential decrease in the number of active switches are given by the MLI cross-connected half-bridges. By expanding the number of module cells or cascading more modules into the inverter, the voltage rating of the CCHB inverter can be easily extended. During startup, it has the advantages of equal capacitor uses and self-balancing. For balancing the voltage capacitor, two-stage voltage balance methods are used. The primary approach is to control feedback dependent upon the proportional-integral, and the second method is to achieve a DC voltage equalization based on the RSS method associated with the level-shifted PWM. The voltage-equalization efficiency was analyzed. Although the proposed approach has been applied to five-level cross-connected half-bridges inverter, it can be extended to any number of voltage steps of CCHB-MLI with relative ease. Furthermore, with minimal steady-state errors, the dynamic response of the voltage balancing is very good. The outcomes of the five-level CCHB-MLI are validated through the simulated and experimental results that achieved from MATLAB model and the laboratory prototypes, respectively. Under the same operating conditions, the experimental waveform is measured and compared to that obtained through simulation.

The majority of this paper is organized in the following way. The structure and operating theory are presented in Section 2. Switching losses and the switching scheme are provided in Section 3. Section 4 addresses the control scheme for the proposed topology. The comparison with conventional cascaded H-bridge topology is contrasted in Section 5. The simulated results achieved in the format of performed in Matlab and Simulink and the experimental results achieved from the actual prototype implementation of the proposed topology, were discussed in Section 6. Eventually, the conclusions from the study are provided in Section 7.

## 2. Five-Level CCHB-MLI

The fundamental structure of a five-level STATCOM-based CCHB-MLI shown in Figure 1. It consists of two DC-capacitors and six active switches. The potentially higher terminal of previous DC-capacitor is linked via active switches to the next DC-link capacitor's lower potential terminal, and vice versa.

Active switches used in the inverter are IGBTs with an antiparallel diode. Each switch has bi-directional current conduction capability, and the capacitors have unidirectional current conduction ability. The active switches are connected in an alternate direction to each other. Between these active switches, the DC-link capacitor is clamped in. The output voltage has five DC-levels, $\pm 2V_{DC}$, $\pm V_{DC}$, and 0. In CCHB-MLI, due to a series of connected capacitors, the voltages are added through power switches. The total number of output voltage steps ($V_{STEPS}$) can be expressed as;

$$V_{STEPS} = 2j + 1. \tag{1}$$

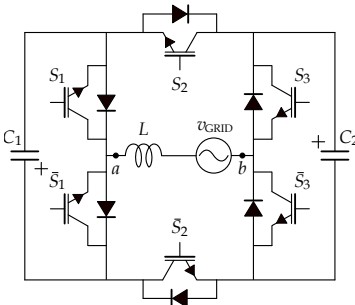

**Figure 1.** Basic structure of five-level cross-connected half-bridges (CCHB) multilevel inverter.

The operating principle of five-level CCHB-MLI is described with $j = 2$ identical DC-link capacitors $C_1 = C_2 = V_{DC}$. There are eight valid operating modes achieved by $j + 1 = 3$ complementary pairs of switches $S_k, \bar{S}_k$ ($k \in \{1, 2, 3\}$), as mentioned in Table 1. Figure 2 indicated the possible output voltage levels and conduction paths. Gating signals $\bar{S}_k$ are generated by inverting $S_k$. The following equation can easily obtain the voltage stress on each switch $S_{VOLT\ STRESS\ (j)}$:

$$S_{VOLT\ STRESS\ (j)} = V_{C(j-1)} + V_{Cj}. \tag{2}$$

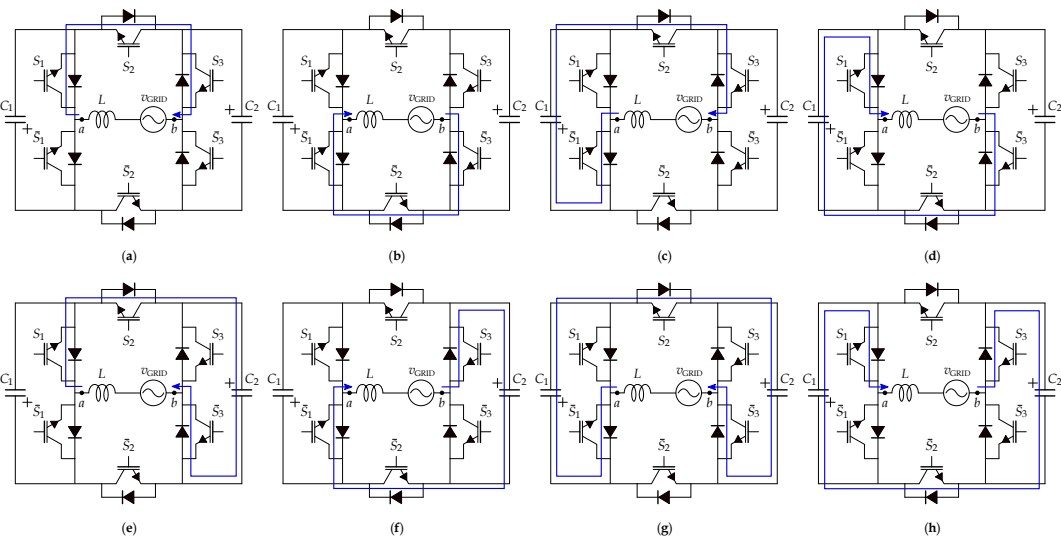

**Figure 2.** Operating and conduction modes of cross-connected half-bridges multilevel inverter (CCHB-MLI): (**a**) Mode 1: $0V_{DC}$. (**b**) Mode 2: $0V_{DC}$. (**c**) Mode 3: $V_{DC}$. (**d**) Mode 4: $-V_{DC}$. (**e**) Mode 5: $V_{DC}$. (**f**) Mode 6: $-V_{DC}$. (**g**) Mode 7: $2V_{DC}$. (**h**) Mode 8: $-2V_{DC}$.

**Table 1.** Switching states of five-level CCHB-MLI.

| Modes | Switching States | | | Power | Effect on $V_{C_j}$ | $v_{INV}$ |
|---|---|---|---|---|---|---|
| | $S_1$ | $S_2$ | $S_3$ | $P$ | $V_{C_1}, V_{C_2}$ | $V_{ab}$ |
| 1 | 1 | 1 | 1 | $P > 0$ | $\downarrow (V_{C_1} + V_{C_2})$ | 0 |
| 2 | 0 | 0 | 0 | $P < 0$ | $\downarrow (V_{C_1} + V_{C_2})$ | 0 |
| 3 | 0 | 1 | 1 | $P > 0$ | $\uparrow V_{C_1}, \downarrow V_{C_2}$ | $V_{DC}$ |
| 4 | 1 | 0 | 0 | $P < 0$ | $\uparrow V_{C_1}, \downarrow V_{C_2}$ | $-V_{DC}$ |
| 5 | 1 | 1 | 0 | $P > 0$ | $\uparrow V_{C_2}, \downarrow V_{C_1}$ | $V_{DC}$ |
| 6 | 0 | 0 | 1 | $P < 0$ | $\uparrow V_{C_2}, \downarrow V_{C_1}$ | $-V_{DC}$ |
| 7 | 0 | 1 | 0 | $P > 0$ | $\uparrow (V_{C_1} + V_{C_2})$ | $2V_{DC}$ |
| 8 | 1 | 0 | 1 | $P < 0$ | $\downarrow (V_{C_1} + V_{C_2})$ | $-2V_{DC}$ |

Due to similar capacitors for voltage rating i.e., $V_{C_1} = V_{C_2} = V_{DC}$, the voltage stresses appearing on the switch pairs $(S_1, \bar{S}_1,)$ and $(S_{k+1}, \bar{S}_{k+1},)$ are equal to $V_{DC}$. The voltage stress on remaining switches becomes equivalent to $2V_{DC}$ each.

The voltage ($v_{ab}$ [V]) and the inverter current ($i_{INV}$ [A]), in relation with DC-link capacitors and the switching function of the CCHB inverter can be obtained as follows:

$$v_{ab} = \sum_{k=1}^{j+1} v_k. \tag{3}$$

Here $v_k$ [V] is nodal voltage:

$$v_k = (-1)^{k+1}(1 - S_k)(V_{C(k)} + V_{C(k-1)}). \tag{4}$$

Combining (3) and (4), we obtain:

$$v_{ab} = \sum_{k=1}^{j+1} \left[ (-1)^{k+1}(1 - S_k)(V_{C(k)} + V_{C(k-1)}) \right] \tag{5}$$

$$i_k = (-1)^{k+1}(S_k - S_{k+1}) \times i_{INV}. \tag{6}$$

## 3. Switch Losses and Switching Scheme

### 3.1. Switch Losses

A significant method for designing an MLI is the estimation of device losses. Each switching system consists of an antiparallel diode power switch inside the CCHB inverter. The direction of the switching mechanism (switch or diode) depends on the inverter's current direction. For instance, if the current direction is from *a* to *b*, in mode 3, the diodes $\bar{S}_1$ and $S_2$ will conduct, while the $S_3$ switch will conduct as shown in Figure 3a. If indeed the current direction is from *b* to *a*, then $\bar{S}_1$ and $S_2$ switches will conduct and $S_3$ diode will conduct as shown in Figure 3b. Analytical analysis of switching, conduct and total losses is performed for the proposed inverter. The system referred to in the power loss calculation in [24], which is based on the extrapolation of the producer's data sheet, is used. SK60GAL123 (Semikron) with a rating of 1200 V, and 50 A is the switching power unit used for the study. For the loss calculation, simulation of current through each system and its data on voltage blocking shall be considered. The total power loss in the MLI is mainly due to losses from switching and losses from conduction. Owing to the delay associated with the transition from ON to OFF, the switching failure is inherent and vice versa, as shown in Figure 4. Overall, the distribution of losses shown in Figure 4a shows that in high-frequency switches, the highest loss occurs. The switching losses in $S_2 - \bar{S}_2$ are minimal since these switches operate at the fundamental frequency, as shown in Figure 4b. Figure 5a depicts the overall switching losses $P_{SW\ LOSSES}$ [W], which can be expressed as follows:

$$P_{SW\ LOSSES} = \frac{1}{T} \sum_{j=1}^{n} (E_{ON(j)} + E_{OFF(j)} + E_{rr(j)}) \tag{7}$$

here, $n$ is the number of transitions in a fundamental period $T$, $E_{\text{ON}}$ and $E_{\text{OFF}}$ are the energy required to turn -ON and -OFF the IGBT, respectively.

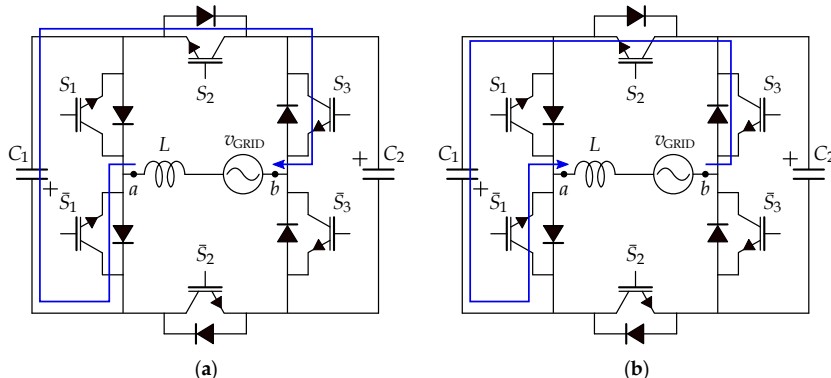

(**a**)                                        (**b**)

**Figure 3.** Conduction path of devices in mode 3 of CCHB-MLI: (**a**) Current direction from $a$ to $b$. (**b**) Current direction from $b$ to $a$.

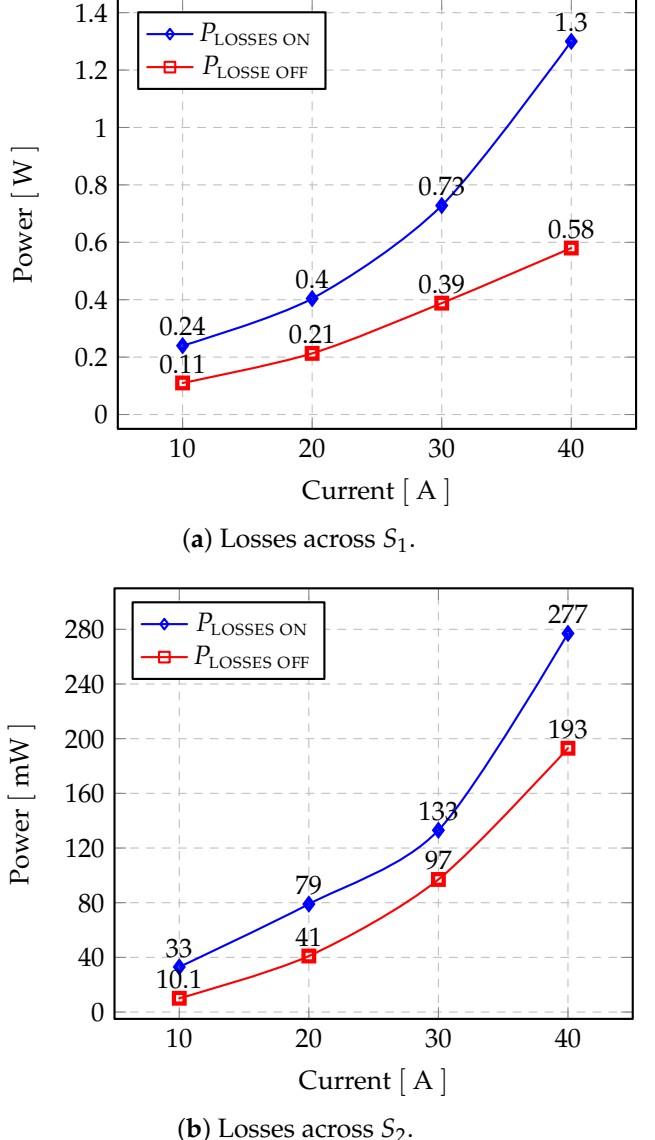

**Figure 4.** ON and OFF states losses of $S_1$ and $S_2$.

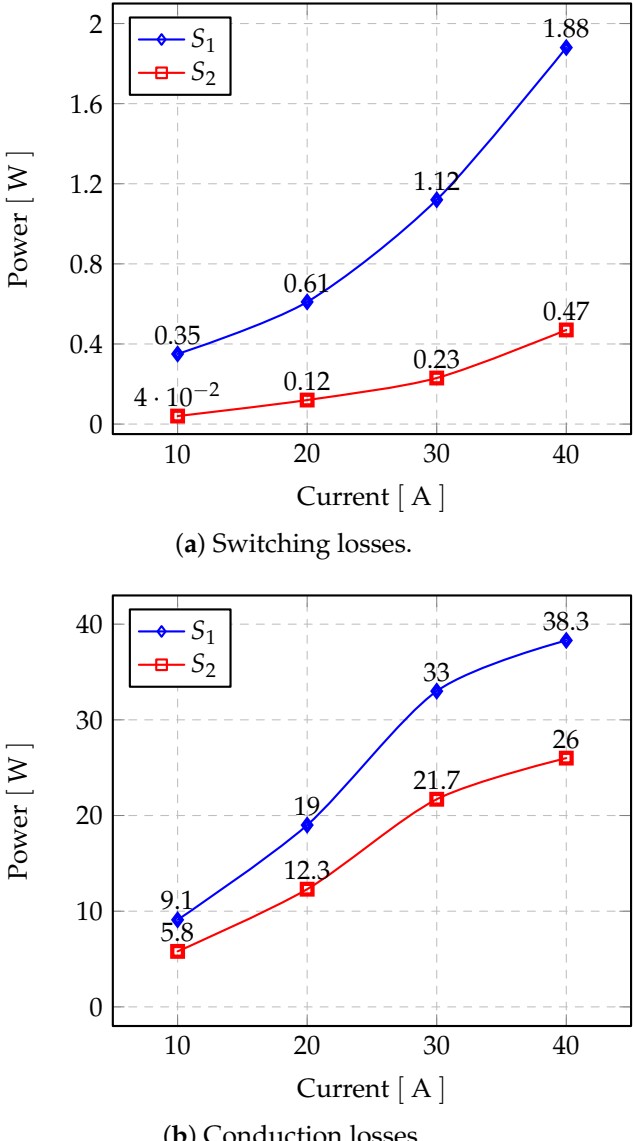

(**a**) Switching losses.

(**b**) Conduction losses.

**Figure 5.** Switching and conduction losses of $S_1$ and $S_2$.

The conduction loss occurs after the switch is ON. The loss of conduction for the ON-state resistance and the forward voltage drop across the switch and body diode. Figure 5b shows the conduction losses $P_{\text{COND LOSSES}}$ [W] of the switching device (IGBT or diode) and can be expressed by the following equation [25]:

$$P_{\text{COND LOSSES}} = \frac{1}{T} \int_0^T (V_F + R_{\text{ON}} \times i_F) i_F dt. \tag{8}$$

Here, $T$ is the time period of the fundamental frequency, $V_F$ is the ON-state forward voltage drop, $R_{\text{ON}}$ is the ON-state resistance, and $i_F$ is the forward current through the device.

Figure 6 shows the total losses ($P_{\text{T}}$[W]) of the inverter, which can be obtained by combining the conduction losses and the switching losses of the switch.

$$P_{\text{T}} = P_{\text{COND LOSSES}} + P_{\text{SW LOSSES}}. \tag{9}$$

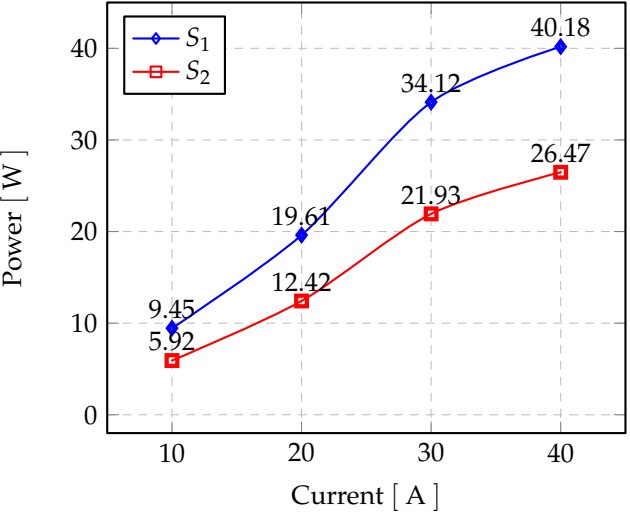

**Figure 6.** Total power losses of $S_1$ and $S_2$.

For STATCOM application, as discussed earlier in Section 1, the inverter is fitted with DC-capacitors. In inductive mode of operation, these capacitors are connected in parallel with the grid. Ripple losses develop the distinction between DC-capacitor voltages and grid voltage. Therefore, the leakage loss $P_{C_{\text{LEAKAGE}}}$ [W] caused by capacitor leakage current $I_{C_{\text{LEAKAGE}}}$ is [26]:

$$P_{C_{\text{LEAKAGE}}} = I_{C_{\text{LEAKAGE}}} \times V_C. \tag{10}$$

The ESR power loss is equal to:

$$P_{\text{ESR}} = I_{C_{\text{RIPPLE}}}^2 \times R_{\text{ESR}}. \tag{11}$$

$$R_{\text{ESR}} = \frac{\tan \phi}{2\pi f_s C}. \tag{12}$$

Here, $R_{\text{ESR}}$ is the capacitor equivalent resistance, and $I_{C(\text{RIPPLE})}$ is the ripple current of the capacitor. $R_{\text{ESR}}$ has the relation Equation (12) with the dissipation factor $\tan \phi$.

The total loss of capacitor is expressed as:

$$P_{C(\text{LOSS})} = P_{C(\text{LEAKAGE})} + P_{\text{ESR}}. \tag{13}$$

The loss of inductors can be calculated in the same manner. However the inductor current $i_L$ is sinusoidal, hence considering the equivalent resistant $R_L$ of the inductor. The loss of the filter inductor $P_{L(\text{LOSS})}$ [W] is:

$$P_{L(\text{LOSS})} = \frac{1}{T} \int_0^T i_{L(\text{RMS})}^2 R_L dt. \tag{14}$$

However, estimates of power losses lead to designing the necessary heat sinks for effective thermal management. The simulated inverter's performance for the different current ratings is shown in Figure 7, while, based on measurements, the efficiency of the developed model is 99.12%.

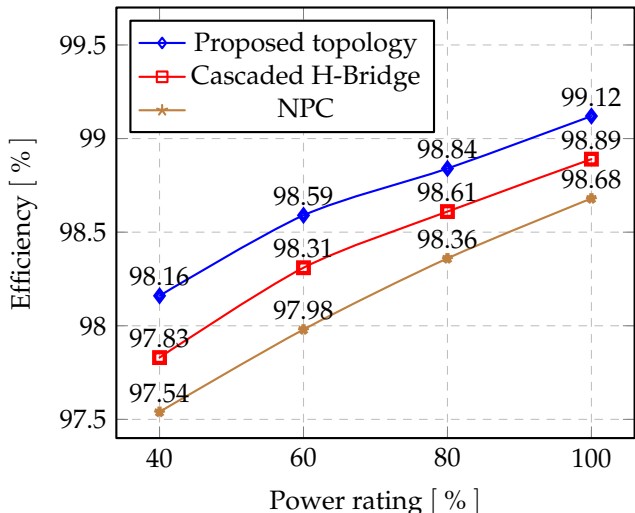

**Figure 7.** Comparison of proposed topology with traditional five-levels topologies.

### 3.2. Proposed Switching Pattern

The suggested topology is based on the modulation technique of the level-shifted carrier. This approach is also spectrally superior to other carrier layouts because, at such unique frequencies, it produces high harmonic concentrations that cancel the output voltage, thus minimizing their overall harmonic distortion. However, if LS-PWM is applied to the CCHB inverter, it is important to link each carrier to a specific cell; otherwise, the capacitor's voltage balancing usual cannot be achieved. This is because the reference signal crosses the carrier at any sampling point and so only the cell connected with the carrier is changed. Consequently, the voltages in the modules will increase or decrease continuously according to the direction of the current $i_{\text{INV}}$, which will differ from their reference values. Therefore, the voltage balance approach is usually slow and relies on the charging conditions. Therefore, to regulate FC voltages at their required dynamic levels, an active balancing method is needed, particularly in transient conditions and unbalanced linear/non-linear loads. Shifted carriers have the same frequency, phase, and magnitude relative to the sine wave reference signal. As shown in the Figure 8, the signal obtained is used for the gating pulse corresponding to the particular voltage levels. The number of level-shifted carriers ($N_{\text{CARRIER}}$) that are required to achieve the required output voltage levels ($V_{\text{LEVEL}}$) is found through the following:

$$N_{\text{CARRIER}} = V_{\text{LEVEL}} - 1. \tag{15}$$

The center switches ($S_2$ and $\bar{S}_2$) are operated by fundamental frequency, while outer switches ($S_1, \bar{S}_1$) and ($S_3, \bar{S}_3$) switches are operated at high switching frequency.

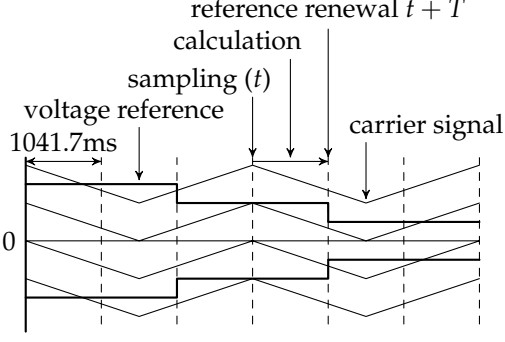

**Figure 8.** Level shift carrier waveform with sinusoidal reference waveform.

## 4. Control Scheme

This section presents the two-stage voltage balancing technique of the cross-connected half-bridges MLI. The PI controller minimizes the average DC voltage error at the first step and the level-shifted modulation technique with the redundant switch state selection (RSS) method implemented at the second stage. In equalizing DC-link capacitor voltages, CCHB-MLI redundant switching modes may play an important role.

Noticing Table 1, mode 1, and 2 are redundant states to obtain a zero-voltage level. To achieve $V_{DC}$, a possible combination of redundant switching states is mode 3 and 5. Similarly, mode 4 and 6 can generate $-V_{DC}$.

The proposed control scheme is divided into three parts,

1. Total capacitor voltage control,
2. STATCOM current control, and
3. Swapping based capacitor voltage control.

### 4.1. Total Capacitor Voltages Control

The goal is to equalize the mean DC voltage to the value relationship of the DC voltage. Comprehensive capacitor voltage control. We follow a simple traditional proportional and integral control system, as shown in Figure 9. The PI controller normalizes the overall the DC-link capacitor error. To originate the total voltage command, the output value is subtracted from the total DC-link capacitor voltage references ($v_C^{REF}$). The instantaneous current reference $i_{INV}^*$ is accomplished from the output of the voltage regulator.

$$i_{INV}^* = \left( v_C^{REF} - \sum_{j=1,2} v_{Cj} \right) \left( k_p + K_i \int dt \right) \sin \theta$$
$$= \Delta v_C \times K_1 \sin \theta \tag{16}$$

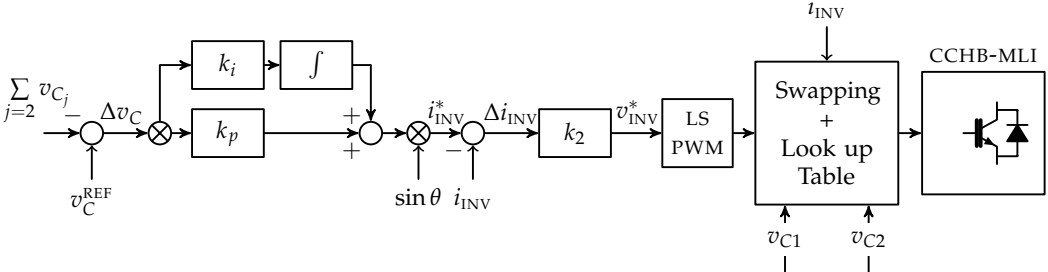

**Figure 9.** Block diagram of voltage and current regulators.

### 4.2. STATCOM Current Control

STATCOM's current control is shown in Figure 9. The current loop control aims to control the reactive power of STATCOM and to control the inverter current. The current STATCOM power is expressed in the following equations in theoretical terms:

$$v_{INV}^* = \left( i_{INV}^* - i_{INV} \right) K_2$$
$$= \Delta i_{INV} \times K_2. \tag{17}$$

Feedback to LS-PWM for the generation of the modulating signal is the output signal. In the next section, the swapping method is suggested and discussed to charge and discharge capacitors equally.

*4.3. Swapping Technique*

Switching DC-link voltage balancing is the last stage of control in this scheme. This control aims to exchange energy to balance the individual capacitor voltage between the two capacitors. By using redundant switching, the $\pm v$ and 0 voltage levels can be achieved in several ways. The modification rule of the DC-link voltages, in short, can be written as follows

- When $(i_{\text{INV}} \times v_{C_1}) < 0$, if $v_{C_1} < v_{C_2}$, then $C_1$ will start charging.
- When $(i_{\text{INV}} \times v_{C_2}) > 0$, if $v_{C_2} < v_{C_1}$, then $C_2$ will start charging.

Using this relation, the $\pm V_{\text{DC}}$ states of the inverter output which can be redundantly selected, is utilized to control the charging or discharging of the DC-capacitors without altering the level shifted PWM.

## 5. Comparative Study

In this section, conventional topologies are compared to the symmetric FC topology. In Table 2, the components needed for different traditional single-phase configuration topologies are shown. Among conventional MLIs, due to sufficient higher-voltage operation without series devices and its modular design, the cascaded H-Bridge MLI topology has been widely used for medium-voltage high-power applications. Therefore, control of DC-link capacitor voltages involves a greater range of voltage sensors, significantly enhancing the cost of the inverter and the STATCOM system's complexities. In this section, a comparative study of CHB-MLI and the proposed topology is discussed. The comparison is performed in terms of the number of devices required, power switches cost, and switch losses. For comparison, both topologies are considered an equal number ($n$) of input voltage sources ($V_{\text{DC}}$). The input voltage sources are symmetric according to voltage rating, $V_{C_1} = V_{C_2} = V_{\text{DC}}$.

**Table 2.** Comparison of traditional multilevel inverters with CCHB.

| Topologies | NPC [27] | CHB [28] | FC [23] | CCHB |
|---|---|---|---|---|
| Main switches | $2(n-1)$ | $2(n-1)$ | $2(n-1)$ | $(n+1)$ |
| Main diodes | $2(n-1)$ | $2(n-1)$ | $2(n-1)$ | $(n+1)$ |
| Clamping diodes | $(n-1)(n-2)$ | 0 | 0 | 0 |
| Dc-sources | $(n-1)$ | $(n-1)/2$ | $(n-1)$ | $(n-1)/2$ |
| Flying capacitors | 0 | 0 | $(n-1)(n-2)/2$ | 0 |
| Dc-sources stress | $(n-1)V_{\text{DC}}$ | $(n-1)V_{\text{DC}}/2$ | $(n-1)V_{\text{DC}}$ | $(n-1)V_{\text{DC}}/2$ |
| PWM scheme | PD-PWM | PS-PWM | PD-PWM | PD-PWM |

In this section, the traditional topologies are compared with symmetric FC topology. The components required of various traditional topologies for single phase configuration are listed in Table 2. Among conventional MLIs, the cascaded H-Bridge MLI topology has been widely utilized for medium voltage high power applications, due to the adequate high operating voltage without series devices and its modular layout. To control DC-link capacitor voltages, a huge number of voltage sensors are needed, which significantly increase the cost of the inverter and the complication of STATCOM system.

*5.1. Component Count*

A major factor in comparing MLI topologies is the number of switches. Not only does a higher range of MLI topologies make it costly and more extensive, but it directly affects the performance and reliability. As a result, fewer switches to MLI topologies have given rise to considerable importance in academia and industry.

To compare switches, the same CHB-MLI output voltage levels and proposed topology are considered. Figure 10 shows the number of switches over the inverter voltage levels. For the

traditional CHB-MLI, each percentage of the appropriate voltage level ($V_{\text{LEVEL}}$) according to the number of switches ($S_n$) is given:

$$V_{\text{LEVEL CHB}} = 2^{S_n/4} + 1. \tag{18}$$

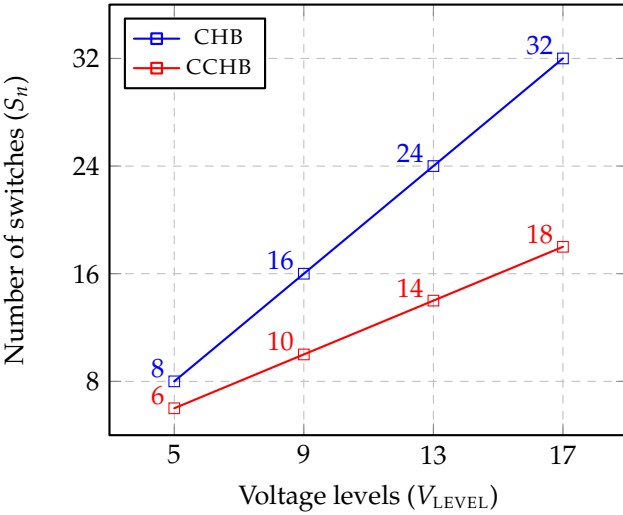

**Figure 10.** Number of required switches for proposed CCHB with CHB topology.

Similarly for proposed topology:

$$V_{\text{LEVEL PT}} = S_n - 1. \tag{19}$$

The gain ($G$) in term of voltage levels ($V_{\text{LEVEL}}$) against number of switches is calculated in the following equation:

$$G_{V_{\text{LEVEL}}} = S_n - 2^{S_n/4}. \tag{20}$$

In percentage, the gain ($G_p$) can be express as:

$$G_p = \frac{S_n - 2^{S_n/4}}{2^{S_n/4} + 1} \times 100. \tag{21}$$

The difference between the number of devices is considerably higher by raising the range of output voltage levels of an inverter in the single and three-phase systems.

### 5.2. Switch Cost

In determining the cost of MLIs, power ratings of switches play the most important role. The current of each switch in the proposed topology is the same as the source current due to the series connection. The different switch voltages are not equal to each other. As a result, relative to different topologies, the total switch voltage blocking can be considered a significant index. One of the most important benefits of the MLI is the low blocking voltage of switches. The proposed topology is composed of high and low blocking voltage switches. If the cost of low blocking voltage switch $V_x$ is $k$ units, then for $2V_x$ voltage rated switch with the same current rating, the cost will be $\xi k$ units, where $\xi$ can be expressed as:

$$\xi = \frac{\text{cost of } 2V_x}{\text{cost of } V_x}. \tag{22}$$

It may be noticed that $\xi$ can vary over a wide range.

The price per unit for the proposed topology (PT) and CHB-MLI can be found as:

$$S_{p\,(\text{CHB})} = 2(V_{\text{LEVEL}} - 1) \times k \tag{23}$$

$$S_{p\,(\text{PT})} = ((V_{\text{LEVEL}} - 3)\xi + 4) \times k. \tag{24}$$

Here, $S_p$ is the total price of the switches and $p$ is the unit price. It should also be remembered that as switches are decreased, the number of gate drives can also be decreased, lowering the actual system's area and weight. Notably, the cost per unit of the proposed and cascaded H-Bridge inverters switches is shown in Figure 11.

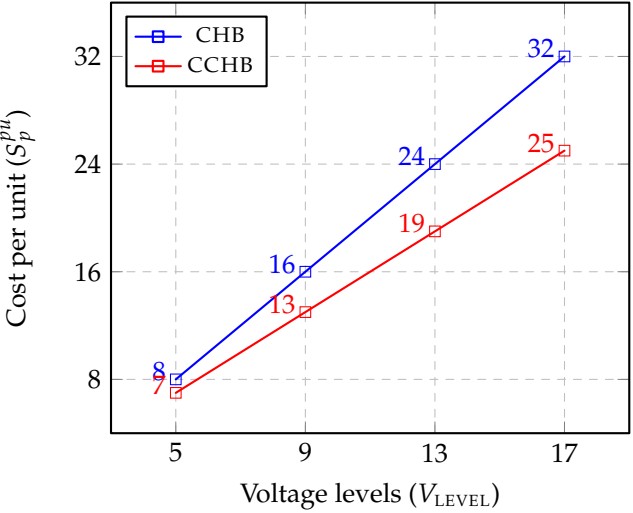

**Figure 11.** Switches costs per unit of proposed CCHB with CHB topology.

### 5.3. Switching Losses

The proposed topology's switching losses in this subsection are contrasted with the cascaded H-bridge for the same degree of the output voltage, as explained previously in Section 3.1. The average switching losses $P_{\text{SW LOSSES}}$ is defined as:

$$P_{\text{SW LOSSES}} = \frac{1}{6}v_{b(j)} \times i \times (t_{\text{ON}} + t_{\text{OFF}})f_j. \tag{25}$$

Let's assume that $\delta = \frac{1}{6}i(t_{\text{ON}} + t_{\text{OFF}})$. Then, Equation (25) is rewritten as:

$$P_{\text{SW LOSSES}} = \delta \times v_{b(j)} \times f_j. \tag{26}$$

Compared to CHB, the proposed topology has fewer switches. All of the eight switches of five-level CHB will operate at a high switching frequency. The power losses are given by:

$$P_{\text{SW\_CHB}} = 8 \times \delta V_{\text{DC}} f_s. \tag{27}$$

The proposed topology (six switches) power losses are defined as:

$$P_{\text{SW(PT)}} = \delta[2(2V_{\text{DC}})f_{\text{LOW}} + 4V_{\text{DC}} \times f_{\text{HIGH}}]. \tag{28}$$

As we know, that four switches in the proposed topology operate with high switching frequency ($f_{\text{HIGH}}$) and it switched at $V_{\text{DC}}$ voltage, while two power switches are controlled by low frequency ($f_{\text{LOW}}$) and switched at $2V_{\text{DC}}$ voltage. Hence:

$$P_{\text{SW(PT)}} = 4 \times \delta V_{\text{DC}}(f_{\text{HIGH}} + f_{\text{LOW}}). \tag{29}$$

Considering that $f_{\text{LOW}}$ is much lower than $f_{\text{HIGH}}$, the switching losses can be equal to:

$$P_{\text{SW(PT)}} \approx 4 \times \delta V_{\text{DC}} \times f_{\text{HIGH}}. \tag{30}$$

From Equations (27) and (30), it is clearly shown that proposed topology switching losses are much lower than CHB, as shown in Figure 7; this is almost half.

## 6. Results and Discussion

Simulation model using MATLAB and simulink and laboratory prototype CCHB-MLI was developed to validate the proposed concept. The voltage balance and current controllers referred to in Figure 9 of Section 4 were simulated and applied. In Table 3, the experimental prototype parameters are shown in Figure 12. There are two sets of findings to illustrate the validity of the proposed topology and controlling strategy. The CCHB-MLI is configured in stand-alone mode at the start to analyze the behavior and output waveforms, and no actual and imaginary forces are transferred to the grid, listed in Figure 13. Constant DC-sources are commonly used alone to achieve a five-level output in the inverter.

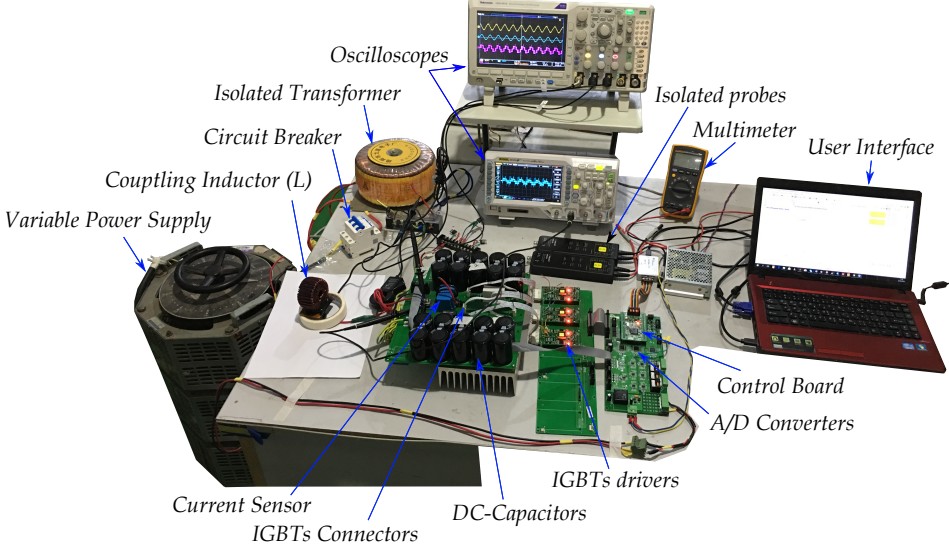

**Figure 12.** Experimental prototype of CCHB multilevel inverter.

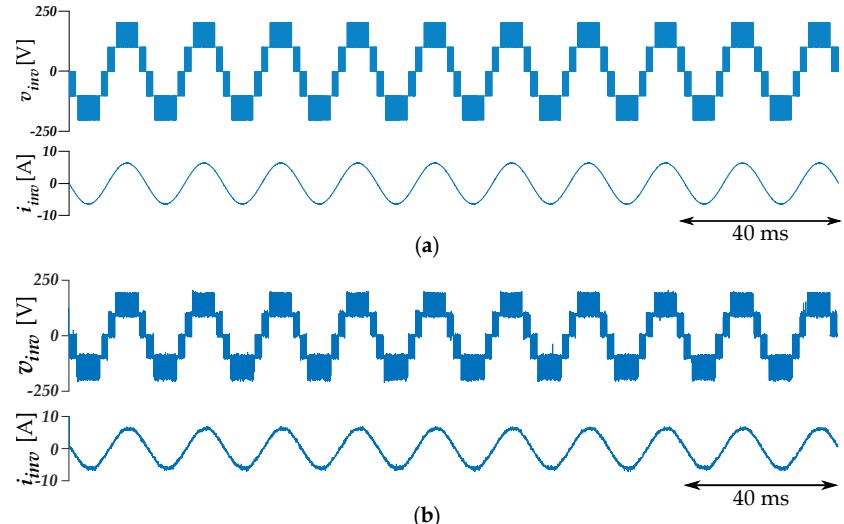

**Figure 13.** Stand alone steady state output voltage and current waveforms with resistive load. (**a**) Simulated waveforms. (**b**) Experimental waveforms.

**Table 3.** System Parameters of CCHB inverter.

| Parameter | Simulation Model | Experimental Prototype |
|---|---|---|
| Inductance | 0.7 mH | 0.7 mH |
| DC-Capacitors | 20 mF | 20 mF |
| Switching Frequency | 3.2 kHz | 3.2 kHz |
| Reference Capacitor Voltage | 100 V | 100 V |
| Reference DC-link Voltage | 200 V | 200 V |
| Grid Voltage (RMS) | 100 V | 100 V |
| $k_i(p)$ | 1 | 0.97 |
| $k_v(p)/k_v(i)$ | 1/0.5 | 1.1/0.7 |
| Control board (DSC) | | TMS320C28346 |
| CPLD | | Altera MAX II (EPM570) |

Consequently, to test the proposed DC-link capacitor voltage balancing technique's effectiveness under transient and stable state conditions, the second part of the results will be carried out in a grid-connected mode. As STATCOM is placed into capacitive operation, Figure 14 shows the output voltages and the STATCOM present. The STATCOM current $i_{\text{INV}}$ phase angle leads the output voltage $v_{\text{INV}}$ by $\pi/2$ rad.

This transient response is mention in Figure 15. Therefore, the STATCOM and the grid voltage-current were in quadrature, indicating a more robust dynamic response to the current loop control.

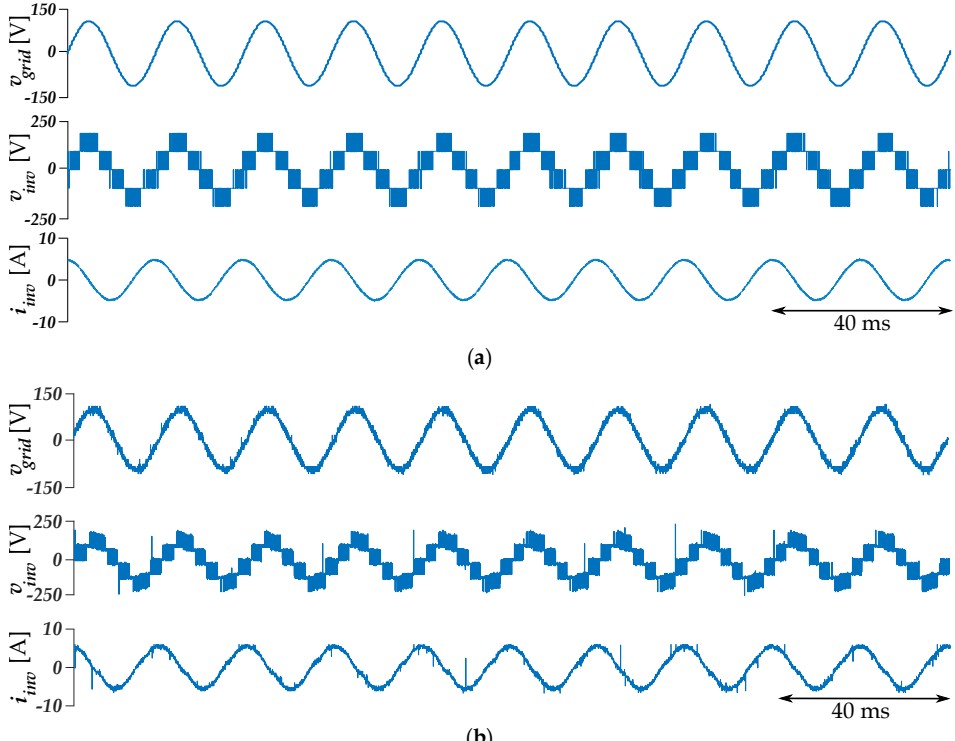

**Figure 14.** Closed loop steady state waveforms of CCHB-MLI. (**a**) Simulated waveforms. (**b**) Experimental waveforms.

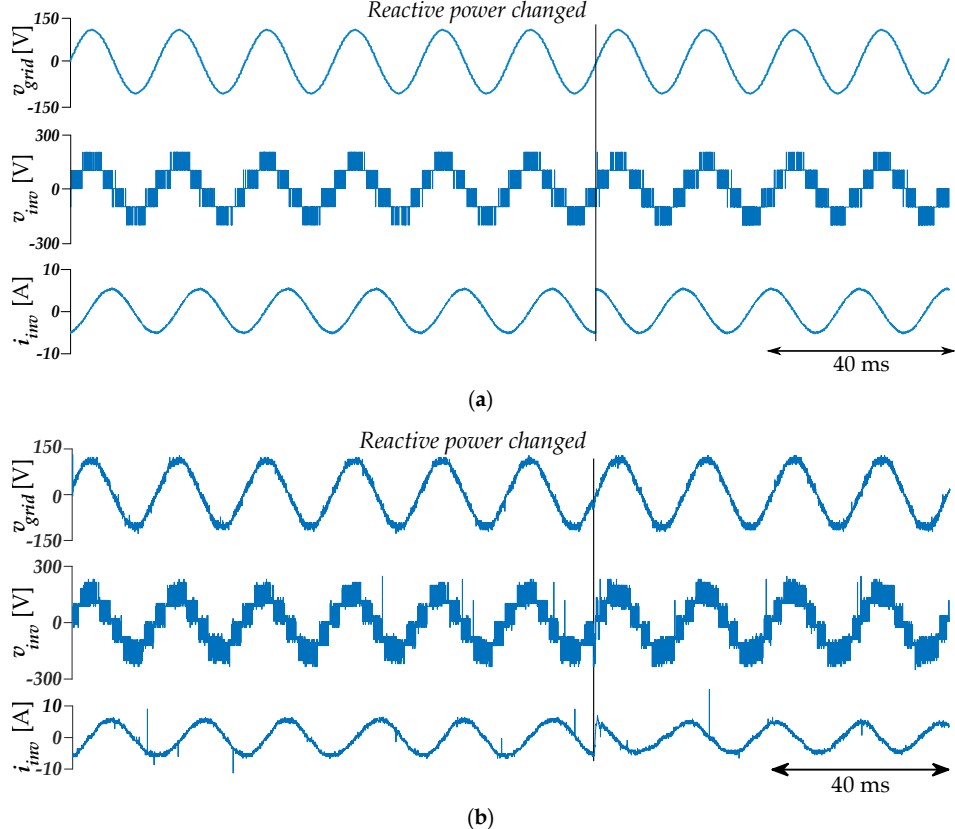

**Figure 15.** Transient state from inductive to capacitive operation. (**a**) Simulated waveforms. (**b**) Experimental waveforms.

The DC voltage waveforms are highlighted in order to observe the validation of the voltage loop regulation. By enforcing the proposed control, DC voltages stay constant toward the reference value within an acceptable range under reactive power changes. A good balance of the capacitor voltage increases the efficiency of the AC-side waveform. Subsequently, to confirm the necessity of this control, the swapping strategy becomes intentionally disabled. The differences are more considerable, and capacitor voltage $V_{C_1}$ starts to increase before enabling the swapping technique, and $V_{C_2}$ begins to decrease, resulting in STATCOM voltage imbalance. The imbalance issues cause distortions in the STATCOM current $i_{INV}$. However, by enabling the swapping algorithm, the problem of divergence has been resolved. Figure 16 evident that a capacitor voltage converges to the reference value, i.e., 100 V in simulation and experimental waveforms. To evaluate the proposed control dynamic response, the imulated load voltage regulation results are performed. STATCOM operates in steady-state mode initially, while no load is attached, as shown in Figure 17. The load is linked to the common coupling point (PCC) after a certain time interval, which dispatch in Figure 17. When operating conditions are changed, STATCOM compensates the current and maintains their respective reference values.

Moreover, the system's dynamic output is also verified, as shown in Figure 18. Initially, there is no reactive power exchange among STATCOM and the utility grid, while it works as a steady-state. At $t = 40$ ms, the reactive load 70.7 $i_{L(RMS)}$ is increased at the point of common coupling (PCC). When the transient occurs, the STATCOM is activated and the load current is compensated over a few cycles. The compensated reactive current becomes stable afterward—see Figure 18. To ensure unit power factor at the load terminals when the grid contribution is zero, STATCOM supplied reactive power to the load. The offset currents will also be updated dynamically following the new reference. The $v_{INV}$ and the compensating current $i_{INV}$ will finally become stable. The findings in Figure 18 show the diverse system output with and without STATCOM.

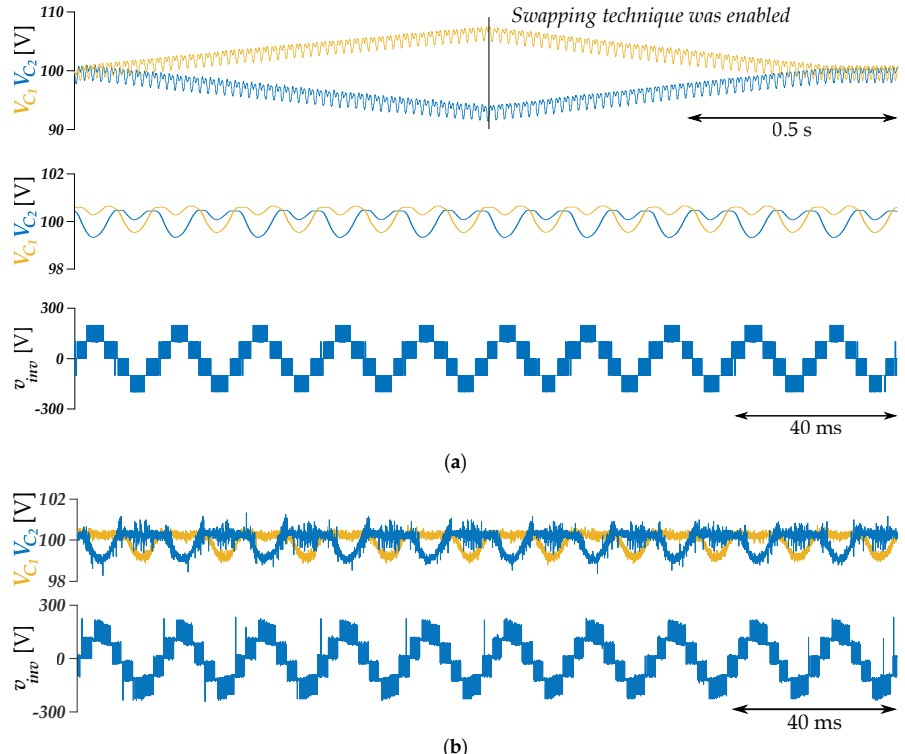

**Figure 16.** Confirming the effectiveness of the total voltage control and swapping technique with capacitor voltage control. (**a**) Simulated waveforms when the total voltage control is active and swapping technique is initially intentionally inactive. (**b**) Experimental waveforms when both total voltage control and capacitor voltage control remain active.

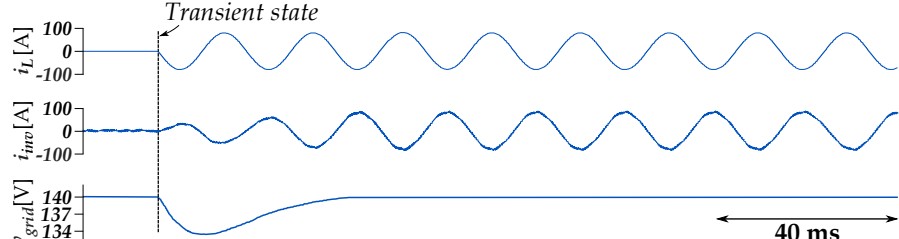

**Figure 17.** Simulated waveforms confirming the compensation effectiveness of load voltage control.

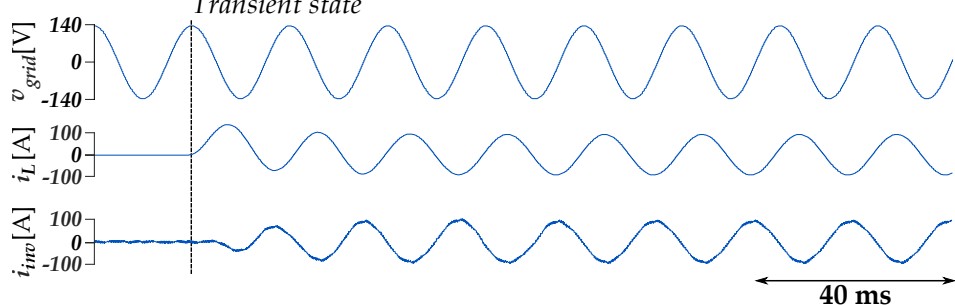

**Figure 18.** Simulated waveforms confirming the dynamic behavior for inductive load compensation.

Figure 19 demonstrates the simulated waveforms when reactive power $q^*$ was increased to 8%, kept constant for 3 s, and again decreased to 8%. Figure 19a shows that STATCOM does not supply reactive power due to intentionally disabled reactive control. The offsets in the line voltage are compensated in several cycles after enabling STATCOM control, as shown in Figure 19b. The proposed control might obtain quick reactive power control without delay time. A variation in the $i^*_{\text{INV}}$ inverter current command is the purpose of such a voltage decrease.

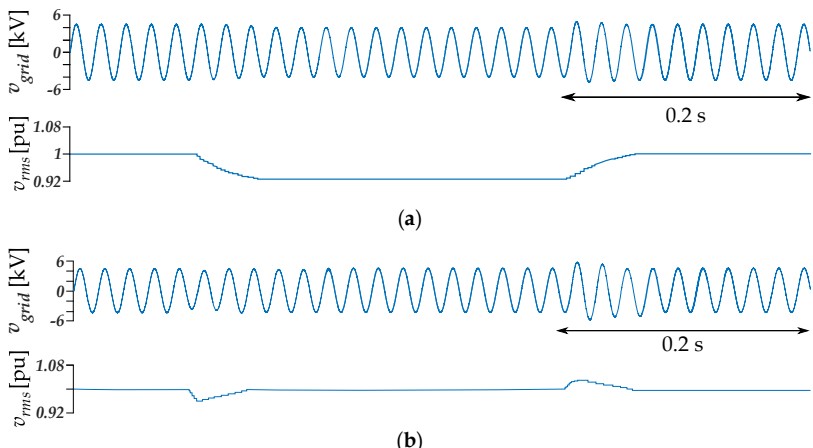

**Figure 19.** Simulated waveforms of voltage sag conditions. (**a**) Without static synchronous compensator (STATCOM). (**b**) With STATCOM.

## 7. Conclusions

In this paper, the CCHB-MLI-based STATCOM is presented. The proposed structure possesses extended capability remarkably. A comprehensive comparison is made between the proposed CCHB-MLI against well-developed topologies regarding their cost, losses, and efficiency. It is noted that the proposed topology has several merits (i.e., reduced switches, volume, and number of gate drivers) over conventional topologies. Active switch loss estimation and switching logic show that the efficiency of the proposed CCHB-MLI is significant for high-power applications. The simulated results for the five-level CCHB-MLI were theoretically predicted and implemented experimentally. With a level-shifted PWM, the proposed dual-loop control makes DC voltages balanced and retained to the reference value. As a consequence, due to asymmetric power losses, aggravated fluctuations and divergence issues are prevented. Finally, the results verified that the CCHB-MLI topology is a good candidate with good performance in the STATCOM topology family.

**Author Contributions:** M.H. proposed the idea for writing the manuscript; Y.L. helped in system parameters and designing to make the simulation and practical test possible and shared the summary of various credible articles to be included in this manuscript. All authors have read and agreed to the published version of the manuscript.

**Funding:** This paper is funded by the State Grid Changzhou Power Supply Company and Changzhou Tianman Energy Technology, Changzhou, Jiangsu, China.

**Conflicts of Interest:** The authors declare no conflict of interest.

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
