# Peer review of "Analysis of Cross-Connected Half-Bridges Multilevel Inverter for STATCOM Application"

_electronics, doi:10.3390/electronics9111898_

Round 1

Reviewer 1 Report

The manuscript entitled: “Analysis of cross-connected half-bridges multilevel Inverter for STATCOM application” proposes a single-phase cross-connected half-bridges multilevel inverter topology for static synchronous compensator applications. It is well written and developed.

Authors tend to group their references. Each article is expected to be provided separately and to be compared with the contribution of this work.

Which are the benefits and the drawbacks of the proposed configuration? How the performance of the proposed system compares with the available alternatives? Is the comparison with other five-levels topologies exhaustive enough? If possible, please further elaborate.

It is of paramount importance that simulating and experimental results are provided to validate the theoretical basis of this work.

Conclusions are supported by the analysis. Having mentioned the above, this manuscript is proposed to be published after only minor revision.

Reviewer 2 Report

This paper studies an innovative multi-level power inverter. In general, the paper is well written and the study is interesting.

All the following indicated aspects should be clarified and better explained in the manuscript.

Paper abstract

  1. The abstract should mention that the conducted experiments are conducted on a prototype.

Introduction / Literature review

  1. The first part of text at pages 1-3 should be included in a section, that could be named as “Introduction”.
  2. The authors should better highlight the innovative aspects of their work in the manuscript.
  3. The authors should describe the applications where the proposed power inverter could be used (e.g., microgrids).
  4. Moreover, the literature review on power inverters reporting the motivations for designing and developing such a kind of topology is lacking. The authors should comment for instance: https://doi.org/10.1016/j.energy.2020.117188, https://doi.org/10.1109/ISGT.2012.6175663, https://doi.org/10.1109/TASE.2020.2986269.

Methodology

  1. The description of the proposed methodology could be deeply improved. First, it could be better to insert at the beginning of the second section (Section 2) an outline about the methodology scheme/architecture (how many steps, the aim of each step, the actors involved in each step, etc.); here, a high-level diagram/scheme could also help reader following the whole description.
  2. All the used variable in all the formula should report the unit.
  3. Is the proposed methodology generic and scalable enough to be applied in different power / voltage range? 
  4. Several scientific studies (for instance: https://doi.org/10.1109/MWSCAS.2015.7282033, https://doi.org/10.1109/SMACD.2015.7301709, documents that could be referenced in the text), show that a viable option for such a kind of closed loop is the phase-lock-loop (PLL) approach. The authors should comment this point.

 Results

  1. To show the worth of the proposed study, the authors could add some comparison with the performances of other state of the art power inverter.

Minor

  1. The authors should check that all the used acronyms are explained (e.g., PWM in the abstract).  

Reviewer 3 Report

This paper presents a dual-loop control technique with level shifted PWM to overcome variations in DC voltages in CCHB MLI with DSTATCOM. A comparison of proposed and existing topologies is presented.

The paper is topic of paper is interesting with some good contributions.

The paper requires some minor revisions before it can be accepted for publication.

  1. The language of the paper requires considerable improvement. 
  2. Authors are advised to revise the abstract and conclusions. Both are bit misplaced and contributions of the paper are not clearly highlighted.
  3. The heading of introduction section is missing.
  4. Table 2 can be enhanced by providing suitable references for other topologies.

Reviewer 4 Report

The submission presents interesting work on the multilevel inverter for the STATCOM application.

The manuscript is well organized with logical structure, according to IMRAD. The title is informative and relevant. The abstract is well written and match the rest of the article. The manuscript introduction section has proper background study, and it is clear what is already known on the application of multilevel inverters; moreover, research is clearly outlined. The drawbacks of the introduction are that section title is missing and abbreviation of multilevel inverter (MLI) defined at the beginning of the manuscript, but not used in the text (lines: 25, 27, 53, 65, 88, etc.).

The study methods used in the manuscript are valid and relevant. The variables are defined and measured appropriately. Tables and figures relevant and clearly presented, and the data shown appropriately.

The conclusions are supported by simulation and empirical results and answer the aims of the study.

I suggest accepting the manuscript to be a part of the MDPI Electronics journal

Round 2

Reviewer 2 Report

In the revised paper several improvements have been added.

Previous comments and concerns have been fully addressed.